# Reduction and Dissolution Behaviour of Manganese Slag in the Ferromanganese Process

**Xiang Li** [1,2] , **Kai Tang** [3] **and Merete Tangstad** [2,*]

1 School of Energy and Power Engineering, Jiangsu University, Zhenjiang 212013, China; xiangli@ujs.edu.cn
2 Department of Materials Science and Engineering, Norwegian University of Science and Technology, 7491 Trondheim, Norway
3 SINTEF Industry, N-7465 Trondheim, Norway; kai.tang@sintef.no
* Correspondence: merete.tangstad@ntnu.no; Tel.: +47-97180255

**Abstract:** The reduction and liquidus behaviour of manganese slag with different basicities were studied in non-isothermal experiments in the temperature range of 1400–1500 °C. Certain amounts of quartz were added to Assmang ore (South Africa), and lime was added to Comilog ore (Gabon), to adjust the charge basicity to 0.5, 0.8 and 1.2. The extent of manganese ore reduction as a function temperature were determined by thermo-gravimetric (TG) balance. Morphology of ores and its change in the course of reduction was examined by scanning electron microscopy (SEM). The results show that the reduction rate of Assmang slag decreases with decreasing basicity, as the liquidus temperature of slag decreases. When spherical MnO phase is present, the activity of MnO is high, and the reduction rate is rapid. Comilog slags show a much higher reduction rate than Assmang slags. The activation energies of MnO reduction between 1400 to 1500 °C are estimated in this study and found to be 230 kJ/mol for Assmang charges and 470 kJ/mol for Comilog charges. The dissolution behaviours of Assmang and Comilog slags were also studied by FactSage simulation and verified by experiments.

**Keywords:** manganese ores; slag; basicity; kinetics; dissolution behaviour

## 1. Introduction

Ferromanganese (FeMn) alloys are essential to iron and steel making because of its sulfur fixing, deoxidizing and alloying properties as well as its low cost. FeMn is commercially produced by carbothermal reduction of manganese ores, primarily in electric submerged arc furnaces (SAF) [1,2]. The ore feed containing higher manganese oxides ($MnO_2$, $Mn_2O_3$ and $Mn_3O_4$) first enters the pre-reduction zone in SAF where water evaporates. The pre-reduction zone involves numerous reactions, as described by following reactions.

Temperature range 25–438 °C:

$$2MnO_2 + CO\ (g) = Mn_2O_3 + CO_2\ (g)$$
$$\Delta G° = -198.79 - 0.0175T\ (kJ).$$

(1)

Temperature range 438–773 °C:

$$3Mn_2O_3 + CO\ (g) = 2Mn_3O_4 + CO_2\ (g)$$
$$\Delta G° = -184.15 + 0.0094T\ (kJ).$$

(2)

Temperature range 773–1200 °C:

$$Mn_3O_4 + CO\ (g) = 3MnO + CO_2\ (g)$$
$$\Delta G° = -56.60 - 0.0327T\ (kJ),$$

(3)

$$Fe_3O_4 + 4CO(g) = 3Fe + 4CO_2(g)$$
$$\Delta G° = -26.82 + 0.0469T\ (kJ).$$

(4)

Carbon dioxide formed by the reduction of high manganese oxides reacts with carbon to regenerate CO through the Boudouard reaction at 800–1200 °C:

$$C + CO_2\ (g) = 2CO\ (g)$$
$$\Delta G° = 120.65 - 0.1728T\ (kJ).$$

(5)

Limestone starts to be decomposed at 900 °C by the reaction:

$$CaCO_3 = CaO + CO_2\ (g)$$
$$\Delta G° = 124.91 - 0.1409T\ (kJ).$$

(6)

In the high temperature zone of the SAF, Mn ores and fluxes melt together, forming the slag, where MnO is reduced to metal by solid carbon (coke or charcoal) or by dissolved carbon in the metal, described by Reaction (7) [3–5]. The correlations of Gibbs free energy change with temperature of Reactions (1–7) are based on the thermodynamic data calculated using HSC Chemistry 7 (Outokumpu, Helsinki, Finland) [6].

$$MnO\ (l) + C\ (s, l) = Mn\ (l) + CO\ (g)$$
$$\Delta G° = 289.54 - 0.1934\ T\ (kJ)$$

(7)

It is widely accepted that this reaction is controlled by the interfacial chemical reaction and the mass-transfer resistance in the slag phase may be neglected [5,7–12]. Daines and Pehlke [11] investigated MnO reduction from basic slags by carbon dissolved in iron, where it was observed that stirring had no effect on the rate. Skjervheim [9] and Tangstad [5] also proved that MnO reduction rate is sensitive to temperature, but not to stirring rate. In the ferromanganese process, the reduction rate of MnO will be rapid when MnO is present in solid state, due to the high and constant activity of MnO. In this stage, because the ore is not completely in the solid stage but consists of a mixture of solid MnO, molten metal and liquid silicate matrix, the relative position of the phases to each other and reduction interface location are dynamic [13–15]. Further reduction of MnO through the homogenous liquid slag phase will take place at a rapidly decreasing rate due to decreasing activity of MnO [5]. In a homogenous liquid slag, higher basicity was found to give higher reduction rate [7–9], which can be explained from the thermodynamic point of view. The activity coefficient of MnO in liquid slag increases with the increase of slag basicity, hence the thermodynamic driving force for Reaction (7) increases.

Different charge materials are likely to exhibit different reduction behaviours during the FeMn process because of their different mechanical and melting properties. In this study, two different manganese ores, Assmang from South Africa and Comilog from Gabon, were used to compare the reduction behaviour during the FeMn process. An industrial SAF operation was simulated by using continuous temperature increase and CO atmosphere in this study.

## 2. Materials and Methods

### 2.1. Materials

The manganese ores used in this study were Assmang and Comilog. Quartz and lime were used to adjust the basicity of the slag system. CO gas with purity of 99.999% was supplied by AGA industrigasser AS (Oslo, Norway). The reductant used in the reaction was metallurgical coke originating from Poland. Note that graphite crucible was used as reaction vessel, which is also considered to be a

carbon source for reduction. All raw materials were crushed and sieved to size range of 0.6–1.6 mm. The chemical compositions of all materials were determined by X-ray fluorescence (XRF) and summarized in Table 1, while the oxidation state of Mn over +2 was determined by titrimetric analysis and given as $MnO_2$, and S was determined by a carbon/sulfur determinator.

**Table 1.** Raw material compositions (dry basis), wt%.

| Components | Comilog | Assmang | Lime | Quartz | Polish Coke |
|:---:|:---:|:---:|:---:|:---:|:---:|
| MnO | 3.63 | 31.4 | | 0.14 | 0.04 |
| $MnO_2$ | 74.0 | 34.8 | | | |
| $Fe_2O_3$ | 3.27 | 15.8 | | | 0.86 |
| $SiO_2$ | 4.71 | 3.75 | 1.00 | 93.9 | 5.60 |
| P | 0.11 | 0.032 | 0.00 | | 0.05 |
| Fix C | | | | | 87.7 |
| $H_2O$ | 4.60 | 0.8 | | | |
| $CO_2$ | 0.16 | 3.7 | 47.50 | | |
| $Al_2O_3$ | 6.97 | 0.29 | 0.27 | 1.19 | 2.79 |
| MgO | 0.03 | 1.31 | 1.00 | 0.05 | 0.22 |
| CaO | 0.06 | 6.4 | 54.00 | 0.09 | 0.42 |
| BaO | 0.2 | 0.75 | | | 0.03 |
| $K_2O$ | 1 | 0.01 | 0.10 | | 0.18 |
| S | 0.009 | 0.21 | 0.011 | | 0.44 |
| Sum, dry basis | 98.8 | 99.3 | 103.9 | 95.4 | 98.3 |

## *2.2. Experimental Procedure*

Non-isothermal reduction experiments were carried out in a laboratory vertical graphite tube furnace equipped with thermo-gravimetric (TG) balance. The experimental conditions are summarized in Table 2. The basicities of original Assmang and Comilog ore used in this study are 2.10 and 0.11, respectively. Neither of them is in the range of industrial ferromanganese slags (0.2–1.2). Certain amounts of quartz were added to Assmang and lime was added to Comilog, to adjust the charge basicities to 0.5, 0.8 and 1.2. For each type of charge, non-isothermal reduction experiments were carried out at temperatures of 1400, 1450 and 1500 °C.

Typically, raw materials were weighed into a graphite crucible with an inner diameter of 30 mm and depth of 61 mm, and mixed well in the crucible by stirring. The usage of raw materials for each experiment can be found in Table 2. The crucible with a lid was suspended by a molybdenum wire to the balance and positioned in the hot zone of the furnace. The crucible temperature was measured by a type B thermocouple, which was ~0.5 cm away from the bottom of crucible. During the whole reduction process, CO flow rate was fixed to 0.5 NL/min. The furnace was heated to 1200 °C with heating rate of 25 °C/min, where it was kept for 30 min. This was to ensure a total pre-reduction of all higher manganese oxides down to MnO. Then, the furnace was heated to the target temperature at a heating rate of 4.5 °C/min. The reduction was stopped by natural cooling in the furnace. The weight loss of sample was recorded every 5 s.

The crucibles containing reacted slag were cast using Epoxy resin and then cut to expose the slag and metal. The samples were ground and carefully polished and coated with carbon film to enhance conductivity during scanning electron microscopy. Electron backscatter images were recorded by field-emission scanning electron microscopy (FESEM, Zeiss Ultra 55 LE, Oberkochen, Germany) operated at 15 kV. The chemical composition of the slag was determined by energy-dispersive X-ray spectrometer (EDS) point analysis.

**Table 2.** Experimental conditions.

| Ore | Temperature °C | Assmang g | Comilog g | Quartz g | Lime g | Coke g | B * |
|---|---|---|---|---|---|---|---|
| Assmang + Quartz | 1400 | 10.0 | | 1.36 | | 2.5 | 0.50 |
| | 1450 | 10.0 | | 1.36 | | 2.5 | 0.50 |
| | 1500 | 10.0 | | 1.36 | | 2.5 | 0.50 |
| Assmang + Quartz | 1400 | 10.0 | | 0.69 | | 2.5 | 0.80 |
| | 1450 | 10.0 | | 0.69 | | 2.5 | 0.80 |
| | 1500 | 10.0 | | 0.69 | | 2.5 | 0.80 |
| Assmang + Quartz | 1400 | 10.0 | | 0.32 | | 2.5 | 1.20 |
| | 1450 | 10.0 | | 0.32 | | 2.5 | 1.20 |
| | 1500 | 10.0 | | 0.32 | | 2.5 | 1.20 |
| Assmang | 1400 | 10.0 | | | | 2.5 | 2.1 |
| | 1450 | 10.0 | | | | 2.5 | 2.1 |
| | 1500 | 10.0 | | | | 2.5 | 2.1 |
| Comilog | 1400 | | 10.0 | | | 2.5 | 0.11 |
| | 1450 | | 10.0 | | | 2.5 | 0.11 |
| | 1500 | | 10.0 | | | 2.5 | 0.11 |
| Comilog + Lime | 1400 | | 10.0 | | 0.84 | 2.5 | 0.50 |
| | 1450 | | 10.0 | | 0.84 | 2.5 | 0.50 |
| | 1500 | | 10.0 | | 0.84 | 2.5 | 0.50 |
| Comilog + Lime | 1400 | | 10.0 | | 1.44 | 2.5 | 0.80 |
| | 1450 | | 10.0 | | 1.44 | 2.5 | 0.80 |
| | 1500 | | 10.0 | | 1.44 | 2.5 | 0.80 |
| Comilog + Lime | 1400 | | 10.0 | | 2.38 | 2.5 | 1.20 |
| | 1450 | | 10.0 | | 2.38 | 2.5 | 1.20 |
| | 1500 | | 10.0 | | 2.38 | 2.5 | 1.20 |

$$* B = \frac{CaO \ wt \ pct + MgO \ wt \ pct + K_2O \ wt \ pct + BaO \ wt \ pct}{SiO_2 \ wt \ pct + Al_2O_3 \ wt \ pct}.$$

## 2.3. Kinetic Calculations

The activity of MnO in the slag remains relatively high as long as solid MnO phase exists in the slag. Previous studies have shown that the reduction rate of MnO can be described by Equation (8) [2,16], assuming that the Reaction (7) in the FeMn slag system is a first-order reaction and controlled by chemical reaction.

$$R_{MnO} = k \cdot A \cdot \left( a_{MnO} - a_{MnO,eq} \right)$$
$$= \underbrace{k_0 \exp(-E/RT)}_{rate \ constant} \cdot \underbrace{A}_{contact \ area} \cdot \underbrace{(a_{MnO} - a_{Mn}p_{CO}/K_T)}_{driving \ force}, \tag{8}$$

where $R_{MnO}$ is the reduction rate (g/min), $k$ is the rate constant (g/min·cm$^2$), $k_0$ is the frequency factor, $A$ is the contact area (cm$^2$), $E$ is the activation energy (kJ/mol), $R$ is the gas constant, $T$ is the temperature (K), $a_{MnO}$ is the activity of MnO in the slag phase, $a_{MnO,eq}$ is the equilibrium activity of MnO in the slag phase, $a_{Mn}$ is the activity of Mn in the metal phase, $p_{CO}$ is the partial pressure of CO (g) and $K_T$ is the equilibrium constant at temperature $T$.

It is assumed that slag–coke contact area is constant during the reduction and that $p_{CO} = 1.0$ as the atmosphere is CO at 1 atm. $K_T$ is obtained from HSC Chemistry 7 [6]. The reduction rate $R_{MnO}$ is calculated by differential of weight loss curves with respect to time.

The $a_{MnO}$ is equal to the mole fraction of MnO in the solid spherical MnO phase if it exists in the slag. If the slag is completely liquid, the $a_{MnO}$ is calculated by Equation (9) [17]. This equation is

obtained by MnO activities calculated by FactSage 7.0 (ThermFact LTD., Montreal, QC, Canada) [18] and linear fitting.

$$
\begin{aligned}
a_{MnO} = C_{MnO} \cdot \exp\big(&0.0007576T - 123.7C_{MnO} + 30.14C_{SiO_2} + 47.84C_{MgO} + 49.54C_{CaO} - \\
&47.96C_{Al_2O_3} + 122.8C_{MnO}^2 - 67.78C_{SiO_2}^2 - 46.32C_{MgO}^2 - 47.68C_{CaO}^2 + 22.51C_{Al_2O_3}^2 + \\
&78.35C_{MnO}C_{CaO} + 77.56C_{MnO}C_{MgO} + 176.6C_{MnO}C_{Al_2O_3} + 101.2C_{MnO}C_{SiO_2} - 71.52C_{SiO_2}C_{CaO} - \\
&70.58C_{SiO_2}C_{MgO} + 27.35C_{SiO_2}C_{Al_2O_3} + 46C_{SiO_2}^3 - 92.97C_{CaO}C_{MgO} + 2.44C_{CaO}^3\big),
\end{aligned}
\tag{9}
$$

where $C_{MnO}$, $C_{SiO_2}$, $C_{CaO}$, $C_{MgO}$ and $C_{Al_2O_3}$ are the mass fractions of MnO, SiO$_2$, CaO, MgO and Al$_2$O$_3$ in the slag phase, respectively. The $C_{MgO}$ is the summation of mass fraction of MgO, BaO and K$_2$O, considering that BaO and K$_2$O in slag behave similarly to MgO.

The $a_{Mn}$ is calculated by the analytical equation, Equation (10), suggested by Tangstad [5]. This equation applies well to FeMn metal in the temperature range of 1300–1600 °C and the whole manganese mole fraction range:

$$
a_{Mn} = X_{Mn} \cdot \exp\big(5.47X_C^2 - 46.8X_C^3 - 13.3X_C X_{Fe} + 17X_C X_{Fe}^2\big),
\tag{10}
$$

where $X_{Mn}$, $X_{Fe}$ and $X_C$ are the mole fractions of Mn, Fe and C in the metal phase, respectively.

The $X_{Mn}$ and $X_{Fe}$ are calculated based on the TG data, assuming that all iron oxide was reduced to Fe metal in the pre-reduction stage and the weight loss after pre-reduction stage is the CO produced from MnO reduction to Mn metal. Assuming that reduced metal is saturated with carbon, the $X_C$ is calculated based on Equation (11) [19]. This equation can be used in in the temperature range of 1300–2400 °C:

$$
\log Cwt\ pct = 0.7005 + 1.12 \times 10^{-4}T + \big(-0.3550 + 0.9910^{-4}T\big)R_{Fe} + \big(0.0294 - 0.082 \times 10^{-4}T\big)R_{Fe}(1 - R_{Fe}).
\tag{11}
$$

where $R_{Fe} = \frac{Fe\ wt\ pct}{Fe\ wt\ pct\ +\ Mn\ wt\ pct}$.

## 3. Results and Discussion

### 3.1. Reduction Behaviour of Assmang and Comilog Ore

Temperature-programmed reduction experiments were carried out in CO atmosphere. The weight loss versus time for different charges is presented in Figure 1, and the temperature profile is also shown. The reduction curve for Assmang with different basicities overlap each other in the pre-reduction zone ($T \leq 1200$ °C). The change of Assmang charge basicity by addition of quartz does not show a visible effect on pre-reduction rate, as expected. The weight loss of Comilog charge (B = 0.11) at the end of pre-reduction stage is higher than that of Assmang (B = 2.10), mainly due to the higher amount of MnO$_2$ content in Comilog ore, as shown in Table 1. The weight loss for Comilog charges looks the same before 900 °C, because they contain same amount of MnO$_2$ and Fe$_2$O$_3$. The basicity has little effect on the reduction of higher manganese oxides and iron oxides, as they are gas–solid reactions. In the temperature range of 900 to 1200 °C, the weight loss for Comilog charges increased significantly with the increase of basicity, due to the weight loss from lime decomposition. Higher basicity Comilog charge contains a greater lime amount, as shown in Table 2, therefore a higher weight loss in the pre-reduction stage is obtained.

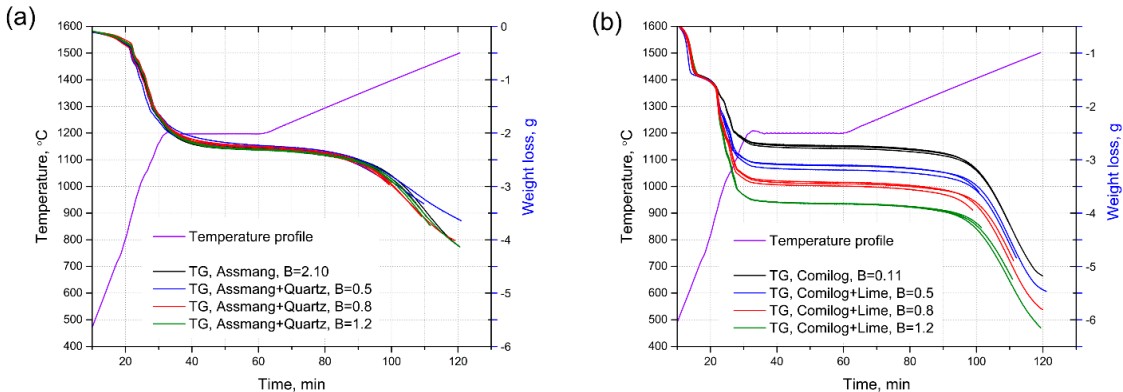

**Figure 1.** Reduction curves of all charges. (**a**) Assmang series; (**b**) Comilog series. Temperature profile is shown.

The weight loss in the MnO reduction stage ($T > 1200\,°C$) is obtained by deducting the pre-reduction weight loss from the total weight loss and presented in Figure 2. The weight loss of original Assmang charge at 1400 °C is higher than original Comilog charge, but lower than the later at 1450 and 1500 °C. This phenomenon also happens at charges with basicity of 0.5, 0.8 and 1.2.

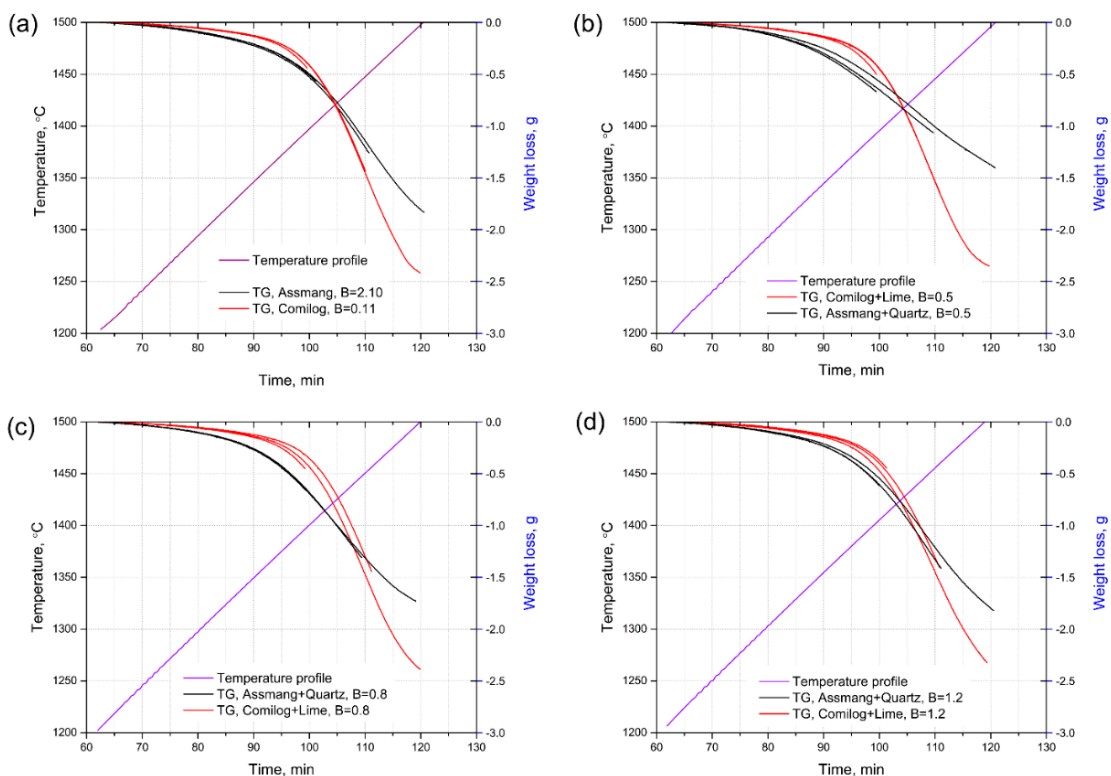

**Figure 2.** Reduction curves of charges with different basicities at MnO reduction stage, weight loss before 1200 °C is deducted. (**a**) Assmang with B = 2.10, Comilog with B = 0.11; (**b**) B = 0.5; (**c**) B = 0.8; (**d**) B = 1.2. Temperature profile is shown.

The MnO reduction rate versus temperature for all charges is calculated by differential of weight loss curve with respect to time, shown in Figure 3. Note that the noise in the curve is in the x-direction (temperature), due to the time corresponding to temperature. The reduction rate will increase with temperature and decrease with lower MnO content, and there are hence two opposing factors with time. All the charges reach the peak reduction rate at around 1450 °C, where the temperature is high and also the MnO content is quite high. For Assmang charges, the reduction rate is increased with the

increase of basicity when *T* > 1425 °C, but the reduction starts sooner for the charges with the lower basicities. There is no visible effect of basicity of Comilog charges to the reduction rate. Basically, the reduction rate is higher for Assmang charges at low temperature but significantly higher for Comilog charges at high temperature.

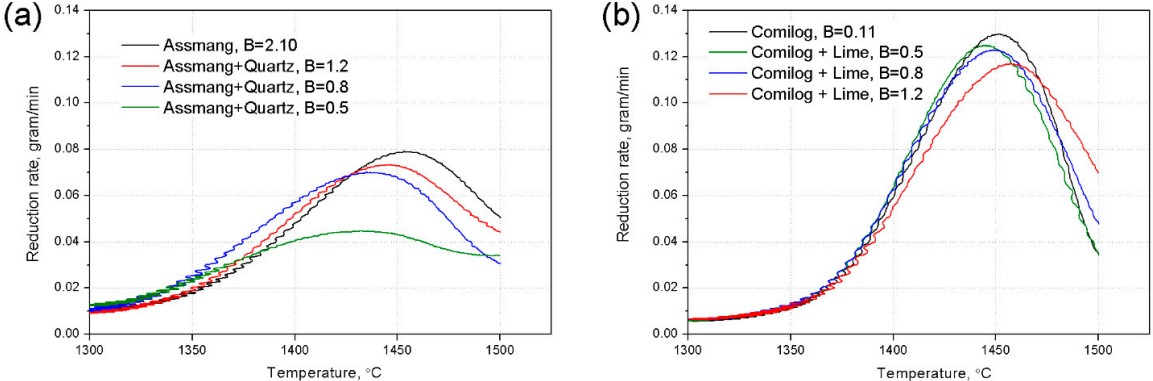

**Figure 3.** Calculated reduction rate versus temperature from all experiments. (**a**) Assmang series; (**b**) Comilog series.

The %MnO in the slag versus temperature is calculated based on the weight loss, presented in Figure 4, assuming that the slag is homogenous. The complete calculated slag compositions are also presented in Table 3. After pre-reduction stage, all Assmang and Comilog charges have a high %MnO in the range of 68.3 to 80.1 wt%. The %MnO decreased slowly before 1400 °C, then dropped dramatically after 1400 °C, indicating that the main MnO reduction stage happens when *T* > 1400 °C. For Assmang charges, the %MnO decreased to 39.5–47.2 wt% at the end of reduction. Among them, the charge with B = 0.5 has the least decrement of %MnO, influenced by the slowest reduction rate. The Comilog charges show a much faster reduction rate when *T* > 1400 °C, compared to Assmang charges, and come to a low range of %MnO at the end of reduction (13.5–21.2 wt%).

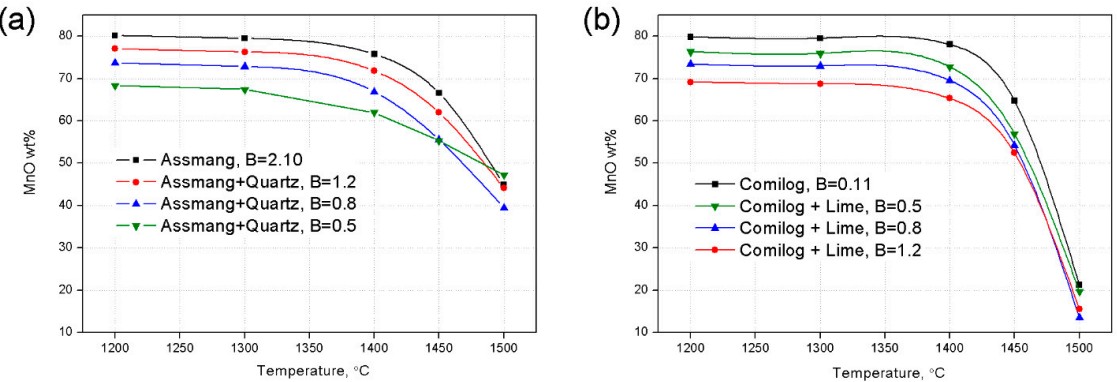

**Figure 4.** Calculated %MnO in the slag versus temperature from all experiments. (**a**) Assmang series; (**b**) Comilog series.

**Table 3.** Calculated slag composition based on weight loss results.

| Ore | T °C | MnO wt% | SiO$_2$ wt% | Cao wt% | MgO wt% | Al$_2$O$_3$ wt% | BaO wt% | K$_2$O wt% | Sum wt% |
|---|---|---|---|---|---|---|---|---|---|
| Assmang B = 0.5 | 1200 | 68.3 | 20.4 | 7.4 | 1.6 | 1.3 | 0.9 | 0.1 | 100.0 |
| | 1300 | 67.4 | 21.0 | 7.7 | 1.6 | 1.4 | 0.9 | 0.1 | 100.0 |
| | 1400 | 62.0 | 24.5 | 8.9 | 1.9 | 1.6 | 1.0 | 0.1 | 100.0 |
| | 1450 | 55.3 | 28.8 | 10.5 | 2.2 | 1.9 | 1.2 | 0.1 | 100.0 |
| | 1500 | 47.2 | 34.0 | 12.4 | 2.6 | 2.2 | 1.4 | 0.1 | 100.0 |
| Assmang + Quartz B = 0.8 | 1200 | 73.7 | 14.3 | 8.0 | 1.7 | 1.3 | 0.9 | 0.1 | 100.0 |
| | 1300 | 72.8 | 14.8 | 8.3 | 1.7 | 1.4 | 1.0 | 0.1 | 100.0 |
| | 1400 | 66.9 | 18.0 | 10.1 | 2.1 | 1.7 | 1.2 | 0.1 | 100.0 |
| | 1450 | 55.5 | 24.1 | 13.5 | 2.8 | 2.2 | 1.6 | 0.1 | 100.0 |
| | 1500 | 39.5 | 32.9 | 18.4 | 3.9 | 3.0 | 2.1 | 0.2 | 100.0 |
| Assmang + Quartz B = 1.2 | 1200 | 77.0 | 10.5 | 8.4 | 1.8 | 1.3 | 1.0 | 0.1 | 100.0 |
| | 1300 | 76.3 | 10.8 | 8.6 | 1.8 | 1.4 | 1.0 | 0.1 | 100.0 |
| | 1400 | 71.8 | 12.8 | 10.3 | 2.2 | 1.6 | 1.2 | 0.1 | 100.0 |
| | 1450 | 61.9 | 17.4 | 13.9 | 2.9 | 2.2 | 1.6 | 0.1 | 100.0 |
| | 1500 | 44.1 | 25.5 | 20.4 | 4.3 | 3.2 | 2.4 | 0.2 | 100.0 |
| Assmang + Quartz B = 2.1 | 1200 | 80.1 | 6.9 | 8.7 | 1.8 | 1.3 | 1 | 0.1 | 100 |
| | 1300 | 79.5 | 7.1 | 9 | 1.9 | 1.4 | 1 | 0.1 | 100 |
| | 1400 | 75.8 | 8.4 | 10.6 | 2.2 | 1.6 | 1.2 | 0.1 | 100 |
| | 1450 | 66.6 | 11.6 | 14.7 | 3.1 | 2.2 | 1.7 | 0.1 | 100 |
| | 1500 | 44.9 | 19.1 | 24.2 | 5.1 | 3.7 | 2.8 | 0.2 | 100 |
| Comilog B = 0.11 | 1200 | 80.9 | 7.7 | 0.2 | 0.1 | 9.7 | 0.0 | 1.3 | 100.0 |
| | 1300 | 80.6 | 7.9 | 0.2 | 0.1 | 9.9 | 0.0 | 1.3 | 100.0 |
| | 1400 | 78.1 | 8.9 | 0.2 | 0.1 | 11.1 | 0.0 | 1.5 | 100.0 |
| | 1450 | 64.7 | 14.3 | 0.4 | 0.2 | 17.9 | 0.1 | 2.4 | 100.0 |
| | 1500 | 21.2 | 31.9 | 0.9 | 0.4 | 40.0 | 0.1 | 5.5 | 100.0 |
| Comilog + Lime B = 0.5 | 1200 | 76.3 | 7.4 | 5.6 | 0.2 | 9.2 | 0.0 | 1.3 | 100.0 |
| | 1300 | 75.9 | 7.5 | 5.7 | 0.2 | 9.3 | 0.0 | 1.3 | 100.0 |
| | 1400 | 72.8 | 8.5 | 6.5 | 0.2 | 10.6 | 0.0 | 1.4 | 100.0 |
| | 1450 | 56.9 | 13.4 | 10.2 | 0.4 | 16.7 | 0.1 | 2.3 | 100.0 |
| | 1500 | 19.7 | 25.1 | 19.0 | 0.7 | 31.1 | 0.1 | 4.3 | 100.0 |
| Comilog + Lime B = 0.8 | 1200 | 73.4 | 7.2 | 9.1 | 0.3 | 8.8 | 0.0 | 1.2 | 100.0 |
| | 1300 | 73.0 | 7.3 | 9.3 | 0.3 | 9.0 | 0.0 | 1.2 | 100.0 |
| | 1400 | 69.6 | 8.2 | 10.4 | 0.3 | 10.1 | 0.0 | 1.4 | 100.0 |
| | 1450 | 54.2 | 12.3 | 15.7 | 0.5 | 15.2 | 0.1 | 2.1 | 100.0 |
| | 1500 | 13.5 | 23.3 | 29.6 | 0.9 | 28.7 | 0.1 | 3.9 | 100.0 |
| Comilog + Lime B = 1.2 | 1200 | 69.2 | 6.9 | 14.1 | 0.3 | 8.4 | 0.0 | 1.2 | 100.0 |
| | 1300 | 68.7 | 7.0 | 14.3 | 0.4 | 8.5 | 0.0 | 1.2 | 100.0 |
| | 1400 | 65.3 | 7.7 | 15.8 | 0.4 | 9.4 | 0.0 | 1.3 | 100.0 |
| | 1450 | 52.4 | 10.6 | 21.7 | 0.5 | 12.9 | 0.0 | 1.8 | 100.0 |
| | 1500 | 15.5 | 18.8 | 38.6 | 1.0 | 22.9 | 0.1 | 3.2 | 100.0 |

The parameters for activation energy calculation at non-isothermal kinetic conditions are summarized in Table 4. The $a_{MnO}$ is equal to the mole fraction of MnO in the solid spherical MnO phase if it exists in the slag, which is much higher compared to when the slag is completely liquid. Figure 5 presents the calculated Arrhenius plots for Assmang (Figure 5a) and Comilog (Figure 5b) charges in the temperature range of 1400–1500 °C. The value of coefficient of determination ($R^2$) is 0.87 for Assmang charges and 0.91 for Comilog charges. The activation energy for Assmang charges is calculated to be 230 kJ/mol, significantly lower than that of Comilog charges (470 kJ/mol). This may explain why Comilog charges have significantly higher reduction rates at high temperature, as the activation energy is the factor showing the temperature dependency [20,21].

**Table 4.** Parameters for kinetic calculation.

| Ore | T °C | Reduction Rate g/min | $a_{MnO}$ | Metal Mole Composition | | | $a_{Mn}$ | $K_T$ | Driving Force | Lnk |
|---|---|---|---|---|---|---|---|---|---|---|
| | | | | Mn | Fe | C | | | | |
| Assmang B = 0.5 | 1400 | 0.040 | 0.96 | 0.39 | 0.38 | 0.24 | 0.16 | 22.3 | 0.95 | −3.16 |
| | 1450 | 0.043 | 0.36 * | 0.48 | 0.27 | 0.25 | 0.18 | 37.7 | 0.35 | −2.10 |
| | 1500 | 0.034 | 0.23 * | 0.53 | 0.21 | 0.26 | 0.20 | 61.8 | 0.23 | −1.88 |
| Assmang + Quartz B = 0.8 | 1400 | 0.057 | 0.90 | 0.41 | 0.35 | 0.24 | 0.16 | 22.3 | 0.90 | −2.75 |
| | 1450 | 0.068 | 0.89 | 0.52 | 0.22 | 0.25 | 0.20 | 37.7 | 0.89 | −2.57 |
| | 1500 | 0.031 | 0.25 * | 0.57 | 0.17 | 0.26 | 0.22 | 61.8 | 0.25 | −2.09 |
| Assmang + Quartz B = 1.2 | 1400 | 0.052 | 0.88 | 0.39 | 0.38 | 0.24 | 0.15 | 22.3 | 0.88 | −2.82 |
| | 1450 | 0.073 | 0.89 | 0.51 | 0.23 | 0.25 | 0.20 | 37.7 | 0.89 | −2.50 |
| | 1500 | 0.044 | 0.84 | 0.56 | 0.17 | 0.26 | 0.22 | 61.8 | 0.83 | −2.93 |
| Assmang + Quartz B = 2.1 | 1400 | 0.048 | 0.94 | 0.38 | 0.39 | 0.23 | 0.15 | 22.3 | 0.93 | −2.96 |
| | 1450 | 0.079 | 0.91 | 0.51 | 0.24 | 0.25 | 0.2 | 37.7 | 0.9 | −2.44 |
| | 1500 | 0.05 | 0.81 | 0.57 | 0.17 | 0.26 | 0.22 | 61.8 | 0.8 | −2.77 |
| Comilog B = 0.11 | 1400 | 0.060 | 0.98 | 0.58 | 0.16 | 0.26 | 0.25 | 22.3 | 0.97 | −2.79 |
| | 1450 | 0.130 | 0.98 | 0.68 | 0.05 | 0.27 | 0.34 | 37.7 | 0.97 | −2.02 |
| | 1500 | 0.034 | 0.05 * | 0.69 | 0.03 | 0.28 | 0.35 | 61.8 | 0.04 | −0.23 |
| Comilog + Lime B = 0.5 | 1400 | 0.064 | 0.98 | 0.59 | 0.16 | 0.26 | 0.25 | 22.3 | 0.97 | −2.72 |
| | 1450 | 0.123 | 0.96 | 0.68 | 0.05 | 0.27 | 0.34 | 37.7 | 0.95 | −2.04 |
| | 1500 | 0.035 | 0.11 * | 0.69 | 0.03 | 0.28 | 0.35 | 61.8 | 0.11 | −1.11 |
| Comilog + Lime B = 0.8 | 1400 | 0.063 | 0.92 | 0.59 | 0.16 | 0.26 | 0.25 | 22.3 | 0.91 | −2.68 |
| | 1450 | 0.123 | 0.91 | 0.68 | 0.05 | 0.27 | 0.34 | 37.7 | 0.90 | −1.99 |
| | 1500 | 0.048 | 0.12 * | 0.69 | 0.03 | 0.28 | 0.35 | 61.8 | 0.11 | −0.84 |
| Comilog + Lime B = 1.2 | 1400 | 0.056 | 0.92 | 0.58 | 0.16 | 0.26 | 0.25 | 22.3 | 0.91 | −2.79 |
| | 1450 | 0.115 | 0.89 | 0.67 | 0.06 | 0.27 | 0.33 | 37.7 | 0.88 | −2.03 |
| | 1500 | 0.070 | 0.80 | 0.69 | 0.03 | 0.28 | 0.35 | 61.8 | 0.79 | −2.43 |

* Calculated based on Equation (9).

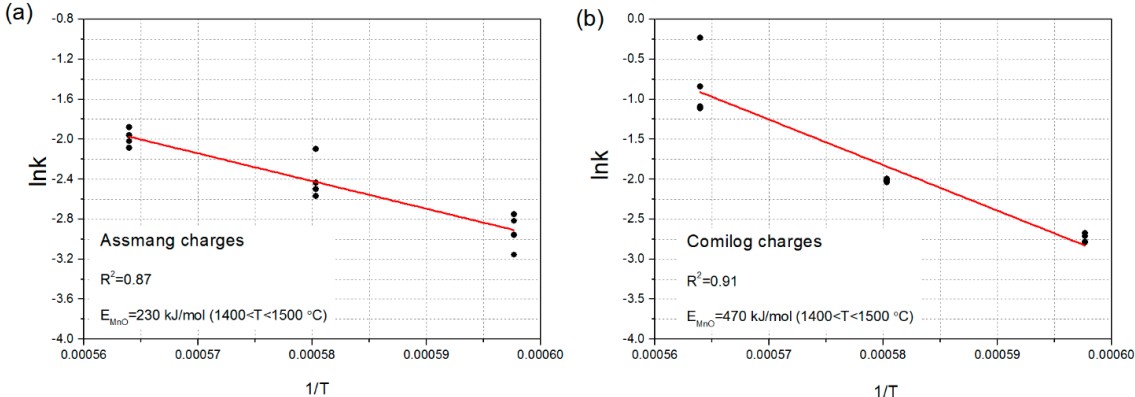

**Figure 5.** Arrhenius plot calculated from reaction rates of reduction for (**a**) Assmang charges; (**b**) Comilog charges from Equation (8).

Table 5 illustrates the reported calculated activation energies for MnO reduction by carbon (solid or dissolved) from different studies. When the MnO reduction is taking place without initial Fe metal, the reported activation energies are in a narrow range (332–407 kJ/mol), which are closed to the free energy of formation of MnO (363 kJ/mol). However, when initial Fe metal is present, a wide range of activation energies (76.7–636 kJ/mol) has been reported. In this study, the initial Fe metal is present when MnO reduction is taking place, because the $Fe_2O_3$ existing in Assmang and Comilog ore was reduced to Fe metal in the pre-reduction stage. The different activation energies of reduction for the two ores may be affected by many factors, such as slag composition, melting behaviour, metal phase

composition and so on [2]. The authors are not sure what causes the different activation energies at the present stage and will study this in the follow-up work.

The influence of viscosity of slag on reduction rate was also considered during this study. However, the viscosity of slag is not easily measured, and the model for determining the viscosity of a $MnO$–$SiO_2$–$CaO$–$Al_2O_3$–$MgO$ system has not been developed or published. A model developed by Tang et al. [22] for a $MnO$–$SiO_2$–$CaO$ system was used to estimate the viscosities of slags in this study at 1500 °C. The estimated viscosities and calculated reduction rates of all charges at 1500 °C are shown in Table 6. The viscosities of different charges varied greatly, however the connection between viscosity and reduction rate is not clear. Previous work also proved that the slag viscosity had little influence on the reduction of $MnO$ [12].

**Table 5.** Reported activation energies for MnO reduction by carbon.

| Reference | Initial Metal | Initial MnO Content wt% | Temperature °C | Calculated Activation Energy kJ/mol |
|---|---|---|---|---|
| [23] | None | 8 | 1550–1700 | 397 |
| [24] | None | 10–70 | 1400–1700 | 375 |
| [9] | None | 35 | 1450–1550 | 407 |
| [5] | None | MnO saturated | 1350–1500 | 347–380 |
| [8] | None | 70 | 1450–1550 | 366 |
| [16] | None | 35 | 1250–1400 | 332–345 |
| [11] | Fe | 4.76 | 1550–1600 | 104 ± 27.2 |
| [7] | Fe | 42 | 1520–1580 | 594, 636 |
| [25] | Fe | 35 | 1500–1600 | 227 |
| [12] | Fe | 59 | 1500 | 250 |
| This study | Fe | 62–78 | 1400–1500 | 230, 470 |

**Table 6.** Parameters for kinetic calculation.

| Ore | Assmang + Quartz B = 0.5 | Assmang + Quartz B = 0.8 | Assmang + Quartz B = 1.2 | Assmang B = 2.1 | Comilog B = 0.11 | Comilog + Lime B = 0.5 | Comilog + Lime B = 0.8 | Comilog + Lime B = 1.2 |
|---|---|---|---|---|---|---|---|---|
| Reduction rate, g/min | 0.034 | 0.031 | 0.044 | 0.050 | 0.034 | 0.034 | 0.048 | 0.070 |
| Iso-viscosity, Poise | 0.75 | 0.75 | Not melted | Not melted | 13.00 | 1.35 | 0.50 | Not melted |

## 3.2. Slag Melting and Phase Development

Figure 6 presents the electron backscatter images of Assmang charges sampled at 1400, 1450 and 1500 °C. All the phases in the images were marked in the images. As an example, for the Assmang charge with B = 1.2 at 1450 °C, it contained a variety of phases. The bright areas are metal prills with main composition of Fe. The slag phase can be divided into two areas: first, an area with spherical $MnO$ particles in a matrix, and second, a crystalline area. The matrix mainly consists of $MnO$, $SiO_2$ and $CaO$. The crystalline area contains MnO-rich dendrites in a matrix. The spherical $MnO$ phase has been in solid stage at the target temperature, while $MnO$ dendrites have been precipitated during cooling. At target temperature the slag would therefore contain a solid $MnO$ phase in coexistence with a liquid phase. For Assmang charge with B = 1.2 and 2.1, the spherical $MnO$ phase exists at all temperatures. When the basicity is reduced to 0.8, spherical $MnO$ phase disappears at 1500 °C, indicating that the slag is completely liquid at 1500 °C. Further lowering of the basicity to 0.5 decreased the liquidus temperature, as spherical $MnO$ phase disappears at 1450 °C.

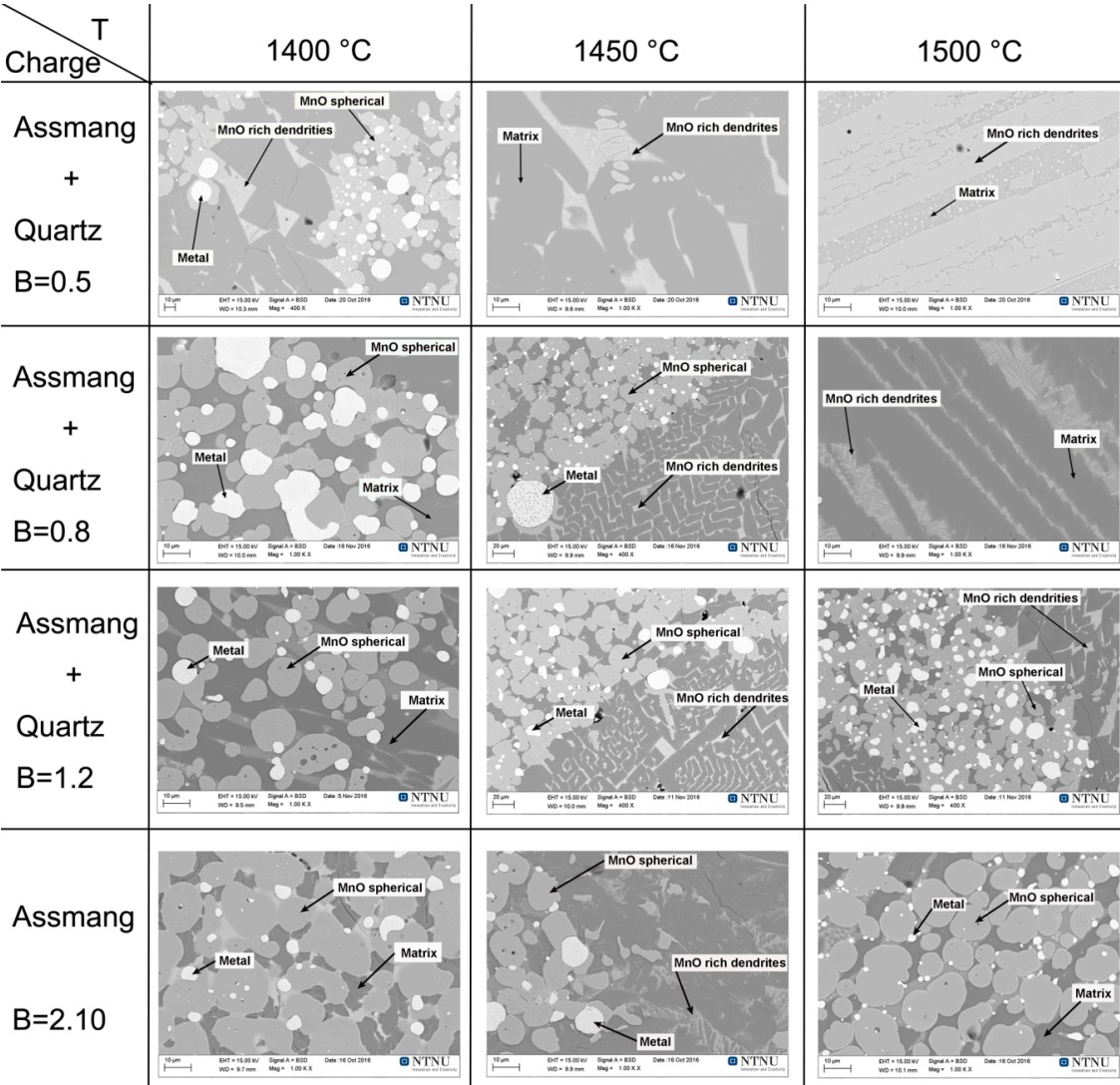

**Figure 6.** Electron backscatter images of Assmang charges with different basicities at different temperatures.

The MnO–SiO$_2$–CaO system is used to simulate Assmang slags, due to the amounts of Al$_2$O$_3$ being insignificant in slag, and other basic oxides (MgO, BaO, and K$_2$O) behaving similarly to CaO [2]. Calculated liquidus relations by FactSage 7.0 with the "FToxid" databases for the MnO–SiO$_2$–CaO system are shown in Figure 7. Green lines represent the reduction paths for Assmang charges with different basicities, red dots represent the starting points after pre-reduction. As the reduction proceeds, the amount of spherical MnO phase decreases, and the composition of charge moves following the reduction path. It shows that Assmang charge with B = 2.10 cannot reach the complete liquid zone in the whole reduction path, in presence of the two stable phases, manganosite + liquid. Assmang charge with B = 1.2 reaches liquidus composition after the %MnO is down to 11.0 wt% at 1500 °C, which is much lower than the experimental %MnO (44.1 wt%). For Assmang charge with B = 0.8, the reduction takes place from 73.7 wt% MnO down to liquidus composition at 52.0 wt% MnO if the temperature is 1500 °C, or at 49.4 wt% MnO if the temperature is 1450 °C. Meanwhile the experimental %MnO is 39.5 wt% at 1500 °C and 55.5 wt% at 1450 °C, which means that the slag reaches liquidus composition between 1450 and 1500 °C. When the basicity of charge decreased to 0.5, the slag goes to complete liquid phase at 59.5 wt% MnO if the temperature is 1400 °C or 61.0 wt% MnO if the temperature is 1450 °C. The experimental %MnO is given at 62.0 wt% at 1400 °C and 55.3 wt% at 1450 °C, indicating that the slag reaches liquidus composition between 1400 and 1450 °C.

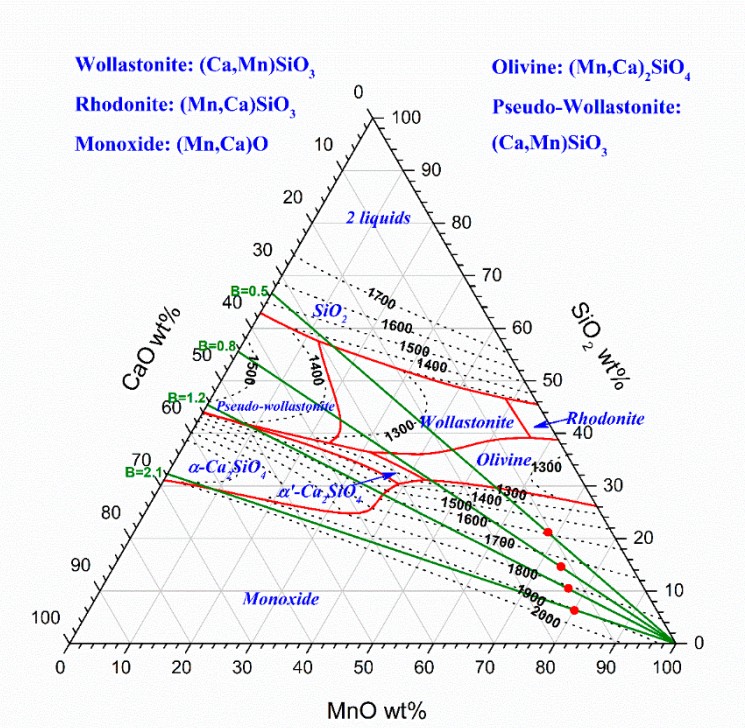

**Figure 7.** Calculated liquidus relations for the MnO–SiO₂–CaO system. Green lines represent the reduction paths for Assmang charges with different basicities, red dots represent the starting points after pre-reduction.

The images shown in Figure 6 indicate that the decrease of Assmang charge basicity by adding quartz promotes the complete melting of slag. According to the liquidus relations shown in Figure 7, the reduction path for Assmang charges with lower basicity flows more easily to the complete liquid zone.

Figure 8 presents the curves of %MnO of the slag at liquidus temperature, calculated from liquidus relations for the MnO–SiO₂–CaO system in Figure 7. It shows that the liquidus %MnO decreases dramatically with increasing basicity of the slag. It also indicates that the liquidus %MnO is less sensitive to temperature in higher basicity slag. The dots in Figure 8 are experimental points, and the (S) marks where spherical MnO phase observed. The dots with (S) are all located in the calculated Liquid + MnO(s) zone. The comparison of theoretical liquidus relations and experimental results shows that phase diagram presented in Figure 8 predicts Assmang charges liquidus behaviour quite accurately.

Figure 9 presents the electron backscatter images of Comilog charges sampled at different temperatures. Calculated liquidus relations for MnO–SiO₂–CaO–Al₂O₃ (Al₂O₃/SiO₂ = 1.24) system are shown in Figure 10. This system is used to simulate Comilog slags. The metal prills are significantly less and smaller than those in Figure 6, because the Comilog ore contains only 3.27 wt% of Fe₂O₃, while Assmang ore contains 15.8 wt% of Fe₂O₃. For the original Comilog ore (B = 0.11), there are some dark cubic crystals in a matrix, except MnO spherical phase and dendrites. EDS analysis results confirm that these cubic crystals are galaxite (MnO·Al₂O₃) phase. At 1450 °C, the reduction will take place from about 80 wt% MnO in presence of two stable phases, manganosite + liquid slag, for example, down to the liquid composition at 62 wt% MnO. From here, further reduction takes place through the galaxite area down to the final composition. When the basicity increases to 0.5 and 0.8, spherical MnO phase disappears at 1500 °C. In the slag there is a presence of MnO dendrites and a dark phase in the matrix. The dark phase is gehlenite (2CaO·Al₂O₃·SiO₂), as confirmed by EDS analysis and also predicted in Figure 10.

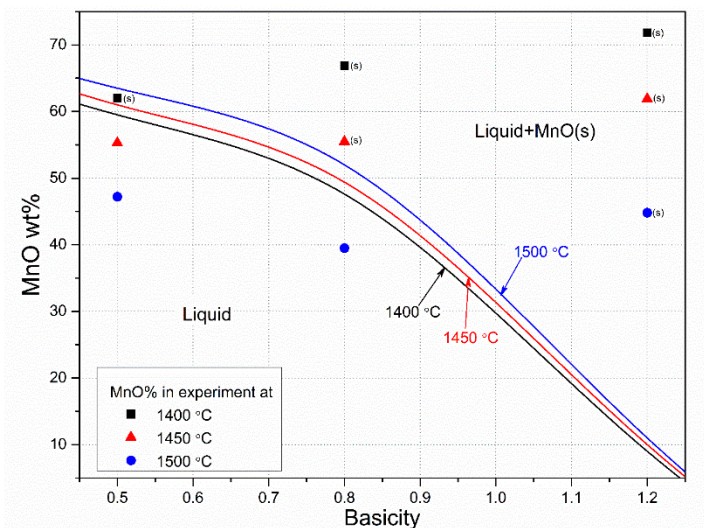

**Figure 8.** Curves of %MnO of the slag at liquidus temperature, calculated from liquidus relations for the MnO–SiO$_2$–CaO system in Figure 7. The dots are experimental points, and the (S) marks where spherical MnO phase is observed.

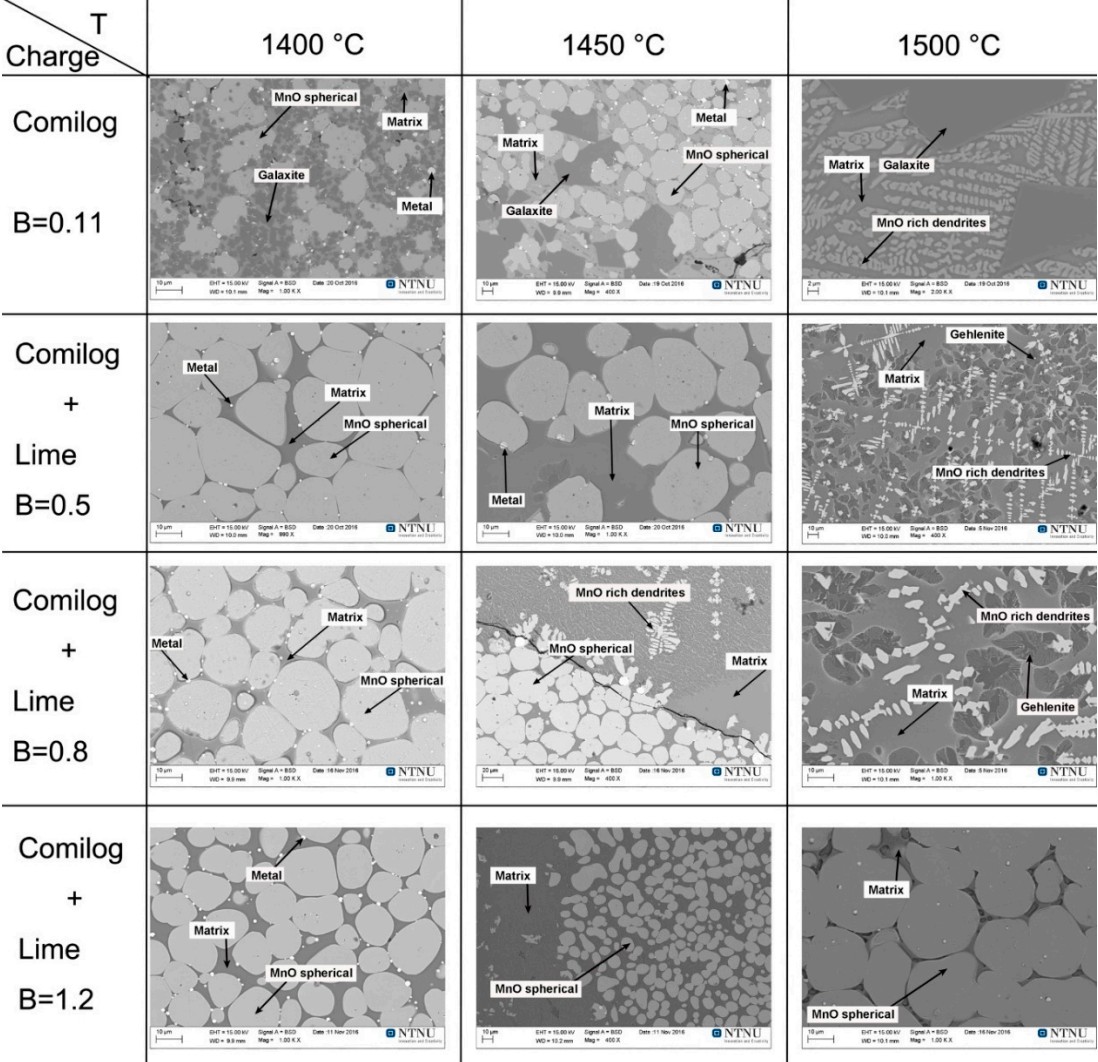

**Figure 9.** Electron backscatter images of Comilog charges with different basicities at different temperatures.

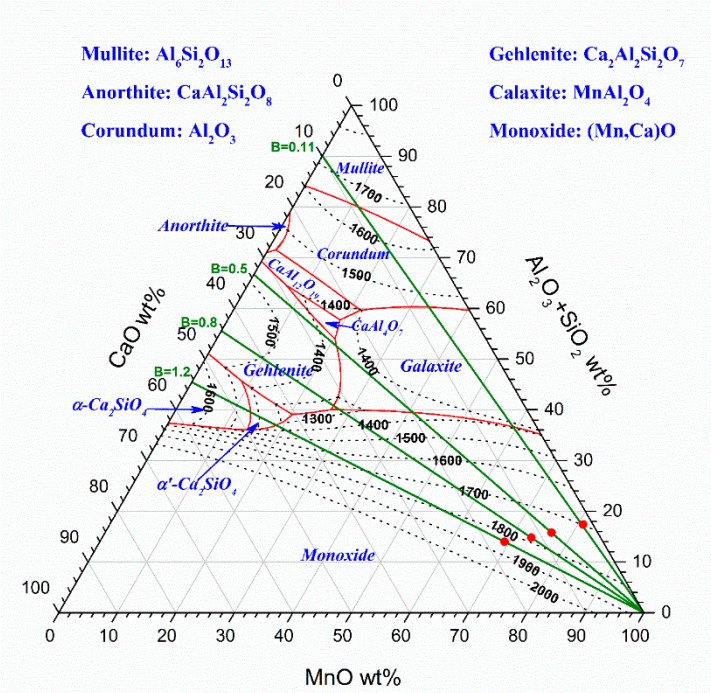

**Figure 10.** Calculated liquidus relations for MnO–SiO$_2$–CaO–Al$_2$O$_3$ (Al$_2$O$_3$/SiO$_2$ = 1.24) system. Green lines represent the reduction paths for Comilog charges with different basicities, red dots represent the starting points after pre-reduction.

Figure 11 presents the curves of %MnO of the slag at liquidus temperature, calculated from liquidus relations for MnO–SiO$_2$–CaO–Al$_2$O$_3$ (Al$_2$O$_3$/SiO$_2$ = 1.24) in Figure 10. The dots in Figure 11 are experimental points, and the (S) marks where spherical MnO phase observed. The dots with (S) are all located in the calculated liquid + MnO(s) zone, except the dot for Comilog charge with B = 1.2 at 1500 °C, which indicates that the phase diagram presented in Figure 10 can predict liquidus behaviour of Comilog charges with low basicities (B = 0.1, 0.5 and 0.8) accurately, but not for the charge with high basicity (B = 1.2).

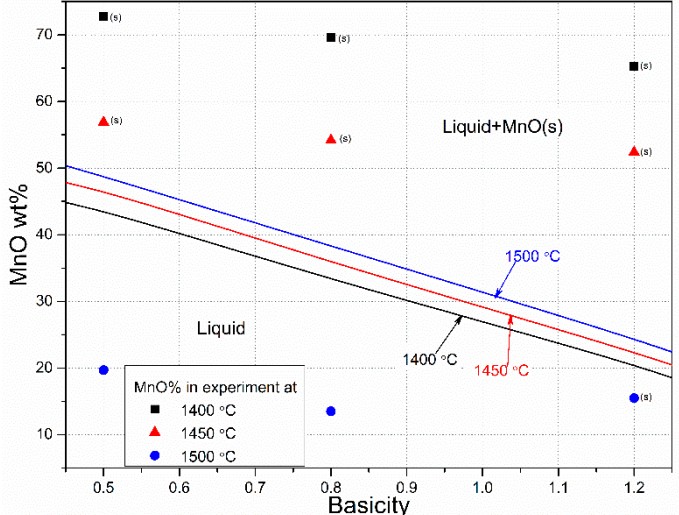

**Figure 11.** Curves of %MnO of the slag at liquidus temperature, calculated from liquidus relations for MnO–SiO$_2$–CaO–Al$_2$O$_3$ (Al$_2$O$_3$/SiO$_2$ = 1.24) system in Figure 10. The dots are experimental points, and the (S) marks where spherical MnO phase is observed.

## 4. Conclusions

The reduction and liquidus behaviour of Assmang and Comilog slags with different basicities are studied in the FeMn process, with a continuous temperature increase under CO atmospheric pressure. The results show that the reduction rate of Assmang charge decreases with decreasing basicity. Comilog charges show a much higher reduction rate than Assmang charges at high temperature, due to a higher activation energy for the Comilog ore. The basicity did not affect the reactivity to a large extent, with the exception of the Assmang ore with a basicity of 0.5. The liquidus relations for $MnO$–$SiO_2$–$CaO$ system and $MnO$–$SiO_2$–$CaO$–$Al_2O_3$ ($Al_2O_3/SiO_2$ = 1.24) systems were calculated and applied to predict Assmang slag and Comilog slag dissolution behaviour, respectively.

**Author Contributions:** Experiment, Data Analysis and Original Draft Preparation, X.L.; Writing—Review & Editing, K.T. and M.T. All authors have read and agreed to the published version of the manuscript.

**Funding:** This research was supported under the Norwegian Research Council (SFI-Metal Production, project No. 90037738), Jiangsu University (High-level Scientific Research Foundation for the introduction of talent, project No. 18JDG015), and Taizhou Institute for New Energy Research, Jiangsu University (project No. 2018-02).

**Acknowledgments:** The authors gratefully acknowledge support from the Norwegian Research Council (SFI-Metal Production, project No. 90037738), Jiangsu University (High-level Scientific Research Foundation for the introduction of talent, project No. 18JDG015), and Taizhou Institute for New Energy Research, Jiangsu University (project No. 2018-02).

**Conflicts of Interest:** The authors declare no conflict of interest.

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
