# Peer review of "Reduction and Dissolution Behaviour of Manganese Slag in the Ferromanganese Process"

_minerals, doi:10.3390/min10020097_

Round 1

Reviewer 1 Report

The author's changes to the manuscript have largely covered the questions that I had previously. There are a few more minor comments:

In Figure 1(b), I was not asking about the large plateau that is obviously related to the limestone content, but the smaller arrest at about 10-15 minutes (~1.5 g mass loss, ~700°C). This appears to be a constant point (independent of basicity). What is causing this arrest? In the reply to Reviewer #1, comment #4, you make good points about the different causes for the different activation energies. It might be useful to also add these points were added to the manuscript. In the reply to Reviewer #2, comment #4, the way you have expressed the explanation appears contradictory. It may be simpler to state that aMnO increases with increasing basicity, and hence the thermodynamic driving force for the reaction increases (especially if you have a reference/model/etc. backing up your argument). For Reviewer #2, comment #12, why did the authors not reword the caption as suggested. The wording as stands is still unclear.

Reviewer 2 Report

The study presented by Li et al. concerns susceptibility to dissolution of manganese slag at temperatures varying from 1400 to 1500C. Overal this is interesting study in terms of experimental and analytical approach. However, my impression out of reading this manuscript was that it is rather a report, in other words I really missed some comparisons with other studies, therefore my first critical point referring to this manuscript would be to improve discussion by developing it via discussion with other literature reports on this topic. Furthermore, I would suggest to elaborate discussion on the data (Fig.6 and 7) and also descriptions referring to figures 6 and 7. Few comments regarding grammar and sentence structure the Authors may find below:

Line 65: please use passive form

Line 71: Just for curiosity, why specific coke from Poland was selected for this study?

Line 86: what was the amount of raw materials used?

Line 144: please rephrase this sentence, what do you mean by room temperatures?

Line 165: please clarify – “due to the time corresponds to temperature”

Line 204: what do the Author mean by “which is acceptable”

Author Response

This manuscript is a resubmission of an earlier submission. The following is a list of the peer review reports and author responses from that submission.

Round 1

Reviewer 1 Report

I have a few specific comments. *Why were the experiments conducted under a CO atmosphere, when according to Eq. 1, CO is a product of carbothermal MnO reduction. *What is the extra step at low temperatures in the weight change curves in Fig. 1(b)? It doesn't appear to be limestone calcination as it is a constant mass change at each basicity. *In Fig. 3, why is the noise in the curve in the x-direction (temperature) rather than in the y-direction (rate). Was the rate calculated from Eq. 2, or from the mass change curves? The last point is probably more significant than the other minor ones: What causes the different activation energies of reduction for the two ores? Different activation energies imply that there are different rate limiting steps or different mechanisms of reduction. Have the authors considered viscosity of the slags at all? If there are large differences in the viscosities of the slags, might mass transfer play a role in determining the rate of reaction?

Reviewer 2 Report

See attached file.
